# Effect of Primary Packaging Material on the Stability Characteristics of Diazepam and Midazolam Parenteral Formulations

**DOI:** 10.3390/pharmaceutics14102061

**Published:** 2022-09-27

**Authors:** María José Rodríguez Fernández, Dolores Remedios Serrano Lopez, Juan José Torrado

**Affiliations:** 1Defense Military Pharmacy Center (CEMILFARDEF), Colmenar Viejo, 28770 Madrid, Spain; 2Quick Identification Laboratory, Military Emergency Unit, Torrejón de Ardoz, 28850 Madrid, Spain; 3Department of Pharmaceutics, School of Pharmacy, Complutense University of Madrid, 28040 Madrid, Spain; 4Institute of Industrial Pharmacy, Complutense University of Madrid, 28040 Madrid, Spain

**Keywords:** diazepam, midazolam, elastomer, stability, primary packing material

## Abstract

Diazepam and midazolam are formulated in autoinjectors for parenteral administration to decrease seizures in the case of emergency. However, the compatibility of these lipophilic drugs with the primary packaging material is a key part of drug formulation development. In this work, diazepam and midazolam were packaged in glass syringes as parenteral solutions using two different elastomeric sealing materials (PH 701/50 C BLACK and 4023/50 GRAY). Syringes were stored at three different storage temperatures: 4, 25, and 40 °C. At different time points over 3 years, physical appearance, benzodiazepine sorption on the sealing elastomeric materials, and drug content in solution were assayed. A detailed study on the adsorption profile of both benzodiazepines on the elastomeric gaskets was performed, indicating that the novel rubber material made of bromobutyl derivatives (4023/50 GRAY) is a better choice for manufacturing autoinjectors due to lower drug adsorption. Diazepam showed a better stability profile than midazolam, with the latter solubilised as a hydrochloride salt in an acidic pH that can affect the integrity of the elastomer over time. The amount of drug adsorbed on the surface of the elastomer was measured by NIR and correlated using chemometric models with the amount retained in the elastomeric gaskets quantified by HPLC.

## 1. Introduction

The quality of parenteral drugs can be affected by their interaction with the materials of the primary packaging. It has been widely reported that diazepam can be adsorbed by plastics contained in its packaging, leading to a drug loss. Parker et al. reported a greater than 24% loss in diazepam potency after packing in polyvinyl chloride (PVC) containers (Viaflex^®^, Travenol laboratories) [1]. Mackichan et al. also reported important losses in diazepam in the plastic tubing during parenteral administration [2]. Cloyd et al. described how diazepam concentration was reduced by about 55% after 2 h of storage in PVC plastic bags and its infusion through plastic administration sets [3]. Parker and MacCara recommended removing the burette chamber in the administration set to reduce the drug-packing material interaction [4]. In 1981, Kowaluk et al. related the loss of drug concentration stored in PVC infusion bags with the lipid solubility of the drug [5]. In 1981, Mason et al. recommended diazepam storage in polyolefin or glass containers but not PVC packing materials because of this interaction [6]. In 1982, Yliruusi et al. also recommended polyethylene and glass containers but not PVC for diazepam parenteral solutions [7]. Martens et al. compared the adsorption of various drugs in PVC containers, and they reported significant losses in diazepam while other drugs, such as midazolam, remained stable [8]. The effect of drug partition coefficients and PVC sorption was modelled by Jenke in 1993 [9]. In 1994, Salomies et al. reported a diazepam loss of 70% in 24 h in contact with PVC, while no loss was detected in polypropylene containers [10]. Tchiakpé et al., in 1995, observed that although lower than PVC, some types of polypropylene materials can also interact with diazepam [11]. Multilayer materials that are a combination of polyethylene, polypropylene, and polyester were proposed in 2005 by Kambia et al. as an alternative to conventional PVC containers to decrease the diazepam sorption [12]. In 2009, Treleano et al. described that the nature of plasticizers was also relevant in the diazepam-PVC interaction [13]. Noh et al. also proposed a new polypropylene multilayer container (Techflex^®^) as an appropriate primary packing container for diazepam [14]. In 2016, Jin et al. proposed polyolefin tubes for diazepam infusion as a better option than polyurethane and PVC tubes [15]. These are just some examples to illustrate the risk of the adsorption of benzodiazepines, especially diazepam, on primary packing materials limiting its availability when administered in vivo.

Benzodiazepines such as diazepam or midazolam are typical components of the acetylcholinesterase inhibiting agents included in autoinjectors for the first treatment of poisoning by chemical warfare agents or pesticides [16,17]. The chemical structure, molecular weight, pKa, and log P characteristics of diazepam and midazolam are described in Table 1. Although these benzodiazepines can reduce the toxicity effects of the nerve agents, their efficacy should be improved by combination with other drugs [18,19,20]. Autoinjection devices are designed to allow intramuscular administration of drugs easily and safely, even for untrained personnel [21]. They are usually disposable, one-dose syringes. Although the syringe for diazepam formulations can be made of glass, the sealing gaskets located at the extremes of the syringe are made of plastic or an elastomeric material (see Figure 1). This point of contact with the drug in the solution can lead to its adsorption over time and, consequently, drug loss. In the paper by Mason et al., similar sealing pieces were removed from the glass syringes after 60 days of storage due to the amount of diazepam adsorbed [6].

The hypothesis underpinning this work is that the material selected for the sealing elastomeric gasket is key to diminishing the interaction with the benzodiazepine and hence, limiting drug loss. No previous work has been performed trying to correlate the amount of drug loss in solution with the drug adsorbed in the primary packing material. The work aims to fabricate autoinjectors for military use containing diazepam and midazolam using two different types of primary packaging materials and to study the stability of both drugs under different conditions. Diazepam and midazolam parenteral formulations were filled in two different types of sealing materials. Drug recovered from the solution and drug adsorbed in the elastomeric material was investigated using different techniques.

## 2. Materials and Methods

### 2.1. Materials

All compounds for formulations were of pharmaceutical grade. Diazepam (≥99% purity) was supplied by Disproquima S.A. (Avd. Del Cerro del Águila 3, San Sebastián de los Reyes, Madrid, Spain) and midazolam (≥99% purity) by Fagron Iberica (ES) (C/ De les Cosidores 150, Terrassa, Barcelona, Spain). Glass syringes were purchased from AGRADO, S.L. (Avd. San Roque 26, Valdemoro, Madrid, Spain). The sealing materials, PH 701/50 C BLACK and 4023/50 GRAY, were purchased from West Pharmaceutical Services Inc. (Exton, PA, USA). All reagents were of analytical grade, and most of them were used as supplied by Fisher Scientific (Madrid, Spain).

### 2.2. Preparation of Benzodiazepine Solutions Loaded into Autoinjectors

The description of formulation composition is reported in Table 2. Formulations were elaborated in classes A, B, and C in cleanroom areas of the pharmaceutical production facilities of the Spanish Army (CEMILFARDEF) in Colmenar Viejo in Madrid (Spain), according to cGMP (EMA, 2021). Both diazepam and midazolam batches formulations were 500 mL quantity, enough to fill 108 syringes for each drug solution.

The diazepam formulation was elaborated by dissolving diazepam (2.5 g) in a mixture of ethanol 96° (42.6 mg) and propylene glycol (233 g). Previously, sodium benzoate (23.75 g), benzoic acid (1.25 g), and benzylic alcohol (7.83 g) were mixed with propylene glycol. Once the diazepam was dissolved, water for injection was added up to 500 mL, and the pH was adjusted to 6.5.

The midazolam formulation was elaborated by dissolving midazolam chloride (2.5 g) in water for injection containing sodium metabisulphite (0.5 g). Once the midazolam was dissolved, water for injection was added up to 500 mL, and the pH was adjusted to 2.9.

The primary packaging of the liquid formulations was a syringe glass Type I (Schott) cartridge of 3 mL capacity filled with 2 mL of the liquid formulation. Half of the syringes were sealed with the elastomeric material “PH 701/50 C BLACK” (coded as JEA, containing chlorobutyl as elastomer type and phenolic resin coating), and the other half of the syringes were sealed with the elastomer “4023/50 GRAY” (coded as JEN, made of bromobutyl material). In Figure 2, both elastomeric materials are illustrated. All syringes were stored protected from light, and placed in a horizontal position for the stability assay.

### 2.3. Stability Study and Specification Limits

Studies were performed according to the ICH Q1A(R2) conditions [22]. Table 3 shows the characteristics of the storage conditions. At each sampling time, three syringes were taken from each storage condition to perform the assay in triplicate. Assays included an evaluation of the physical appearance of the liquid contained in the syringe, the presence of crystals, the appearance of the sealing elastomeric material, benzodiazepine concentration in the liquid contained in the glass syringe, and the amount of benzodiazepine adsorbed in the elastomeric sealing material.

The quality parameters were evaluated, and their specifications were the following: (i) the benzodiazepine concentration in the syringe whose specification is 100 ± 5%, (ii) the amount of benzodiazepine adsorbed in the sealing elastomeric material whose limit is below 5 mg/syringe, and (iii) the appearance of the liquid content and the sealing elastomeric material. The liquid content of the syringe must be pale yellow and transparent without particles. The sealing must retain the initial colour (black or grey, respectively) as well as maintain its flexibility.

### 2.4. Quantification of Benzodiazepines in Solutions and Adsorbed on the Elastomeric Sealing Materials by HPLC

A modular Jasco HPLC equipment with a Jasco PU-1580 pump, a Jasco AS-2050-Plus autosampler fitted to a 100 μL sampling loop, and a UV-visible detector Jasco UV-1575 were used. The wavelength detection was set at 254 nm. The mobile phase was based on the HPLC method described in Eur. Ph. (10th edition, 2019) for diazepam assay consisting of a mixture of acetonitrile (HPLC gradient grade): methanol (HPLC gradient grade): aqueous phase at 22:34:44 (*v:v:v*) proportions. The aqueous phase was a buffer solution of 3.4 g/L KH_2_PO_4_ with pH 5 adjusted with NaOH 2 M. The mobile phase was filtered through a 0.45 µm filter (Supor^®^-450, Pall Corporation Ref 60173) and degassed. The flow rate was fixed at 1 mL/min. The stationary phase was a YMC basic S-C8 (ref. BA99S05-2546WT) column 250 × 4.6 mm with a particle size of 5 µm. The typical working pressure was around 14.4 MPa. The method was previously validated. Linearity was studied between 0.5 and 25 µg/mL. Correlation coefficients were 0.99 for both drugs. Repeatability results showed a variation of 1.2% and 1.9% for diazepam and midazolam, respectively. Reproducibility values were 4.3% and 5.2% for diazepam and midazolam, respectively. The accuracy for diazepam and midazolam at 20 µg/mL were 103.8% and 102.1%, respectively. Intermediate precision (RSD) for 20 µg/mL concentration estimated on three different days for diazepam and midazolam were 6.0 and 2.9, respectively.

Figure 3 shows the chromatogram acquired for each benzodiazepine. Retention times for diazepam and midazolam were 11.9 and 12.7 min, respectively. Test samples of the liquid formulations were diluted with methanol (HPLC gradient grade): purified water (50:50) (*v*/*v*) to get a final theoretical concentration of 20 µg/mL and then filtered through a Millex^®^ HV PVDF Millipore filter of 0.45 μm. The amount of benzodiazepine adsorbed on the sealing elastomeric gaskets was also quantified. The two gaskets from each syringe were soaked in a closed glass container with 20 mL of methanol (HPLC gradient grade): purified water (50:50) (*v*/*v*), mixed in a vortex, and then left for stirring for 48 h. Afterwards, the supernatant was filtered, and the benzodiazepine was quantified by HPLC by the method described above.

### 2.5. Correlation of the Adsorption Process of Diazepam and Midazolam on Elastomeric Gaskets between HPLC and NIR Spectroscopy

A set of 21 JEN and JEA gaskets were immersed in a solution of diazepam or midazolam prepared in the same conditions as the one contained in the autoinjectors. At different time points (15, 30, 60, 120, 180, 240, and 1000 min), three gaskets from each solution were fully dried with tissue paper. The surface of the elastomeric gasket was measured by microNIR (Onsite W, Viavi, MTBrandao España) with a NIR reflectance attachment. Spectra were collected with interleaved scans in the 1660−900 cm^−1^ range. Samples were repositioned between each measurement, and measurements were performed in triplicate for each sample. After NIR measurements, the amount of drug adsorbed on each type of elastomeric material was extracted and quantified by HPLC using the same methodology as described above.

Multivariate data analysis was performed using Unscrambler^®^ X software (CAMO Software, Oslo, Norway) [23]. Pre-processing transformation (data normalisation) was used, followed by the second derivative Savitzky-Golay (7 points). Partial Least Squares Regression (PLSR) was used to create calibration models for quantitative analysis of both benzodiazepines on the surface of the elastomeric material. Each spectroscopic signal was correlated with the total content of benzodiazepine adsorbed on the gaskets. The Kernel and NIPALS algorithms were used to compute the estimated regression coefficients for PLSR. The performance of the models was evaluated using the correlation coefficient (R^2^), and the root mean square error (RMSE) to estimate the fit of validation and calibration samples [24]. The full spectra were included in the PLSR for the JEN gaskets, while in the JEA gaskets, only the range within 1120–1250 cm^−1^ was included in the model. In the JEA gaskets, the data points at 15, 30, and 60 min were removed from the chemometric model due to the resin release to the media at earlier time points.

### 2.6. Wettability Studies and Contact Angle Measurement

The contact angle was measured on the flat surface of each type of gasket using deionized water. The sessile drop method was used to calculate the contact angle based on the Young–Laplace equation. The slope of the tangent to the drop at the liquid–solid–vapour interface line was calculated. All measurements were performed in triplicate under ambient conditions of 25 ± 2 °C [25]. Measurements were taken with Image J software (version 1.52k, Wayne Rasband, National Institute of Health, Bethesda, MD, USA) Pendent_drop plugin), which is a public domain, Java-based image processing software package [26].

### 2.7. Imaging

The surface of the elastomeric sealing at time 0 and after 36 months of exposition to benzodiazepine solutions was imaged using a 9MP 2–200× digital microscope (Conrad Electronics, Hirschau, Germany). Images were processed by ImageJ v1.46 image analysis software. The physical appearance of the elastomeric gaskets was evaluated along all the stability studies. Those gaskets whose appearance was visually altered were imaged using a 14 Mpixels camera from an iPhone 12 (Apple, Cupertino, CA, USA).

## 3. Results and Discussion

### 3.1. Physico-Chemical Evaluation

Diazepam solutions stored at 40 and 25 °C changed from an initial pale yellow colour to a darker yellow. This change did not happen in diazepam samples stored at 4 °C or in the midazolam solutions.

In Table 4, the visual appearance of the different elastomeric gaskets in contact with the diazepam and midazolam solutions at different time points under different stress conditions is illustrated. The deterioration of the sealing pieces 4023/50 GRAY (JEN) in contact with the midazolam solution was visually detected by the naked eye compared to the black elastomers (JEA). However, in Figure 4, a morphological alteration in both types of elastomeric materials, JEN and JEA, is noticeable, especially when in contact with the midazolam solution. In the case of the JEA, the surface of the gasket changed from smooth to highly rugous morphology, while the JEN showed a more porous structure at time zero that faded over time in contact with the midazolam solution. Elastomeric gaskets in contact with the diazepam solution showed a less aggressive alteration, which can be related to the lowest acidic pH of the midazolam solution compared to the diazepam one.

### 3.2. Benzodiazepine Stability in Solution and Recovered from the Elastomeric Gaskets

Figure 5 shows the evolution of benzodiazepine’s recovery under different storage conditions based on the different types of elastomer gasket used. The temperature had a significant and direct effect on the loss of both benzodiazepines (*p*-value < 0.01, ANOVA). In our experimental conditions, diazepam formulations were significantly (*p*-value < 0.001, Student’s t-test) more stable than midazolam formulations. The effect of temperature and storage time on the degradation of diazepam was also described by Smith et al. [27]. Diazepam and midazolam were stable (>90% drug content) at 4 °C over 12 months. However, drug loss was significantly higher at 25 °C when the JEA elastomeric gaskets were utilised. This was more noticeable with midazolam, whose drug content went well below 90% after 3 months of storage in the presence of JEA gaskets. At 40 °C, drug degradation was more pronounced. However, no significant differences were observed between JEA and JEN for the midazolam. This can be related to a deep alteration in the integrity of the elastomeric gaskets at this temperature, resulting in similar drug adsorption. In fact, drug loss in the solution is probably due to the combined effect of drug sorption on the elastomeric material and chemical drug degradation in the solution.

This combined effect was studied in more detail in Figure 6. The decrease in drug loss is inversely correlated with an increment of drug adsorbed in the elastomeric material. This highlights the importance of preventing or minimising drug adsorption during storage of the autoinjectors to comply with Pharmacopeia specifications. Black elastomeric gaskets failed to provide one-year midazolam stability at 25 °C, while grey elastomeric gaskets provided at least 18 months of stability. Bearing in mind the overall role that played the drug adsorption on the elastomer gasket, this process was studied further in a separate study.

### 3.3. Evaluation of the Adsorption Process of Benzodiazepines by HPLC and NIR Chemometric Analysis

Figure 7 shows the adsorption process of diazepam and midazolam on the two different elastomeric materials tested, GRAY JEN and BLACK JEA. Diazepam showed about 2-fold greater adsorption (*p*-value > 0.05) on the BLACK gaskets compared to midazolam. Further, both benzodiazepines showed significantly higher adsorption at earlier times on BLACK JEA gaskets (509 and 200 µg of diazepam and midazolam, respectively, per gram of elastomer) compared to GRAY JEN materials (299 and 36 µg of diazepam and midazolam, respectively, per gram of elastomer). This is the first report that provides evidence about the use of novel JEN elastomeric materials as a better choice than conventional BLACK JEA gaskets when formulating autoinjectors loaded with benzodiazepines. Other authors have reported an enhanced apparent degradation rate of interferon in contact with cholobutyl stoppers compared to I borosilicate ampoules, which, in our opinion, can be related to a poor drug recovery from the elastomeric stopper rather than chemical degradation in the solution [28].

This phenomenon can be related to the different hydrophobicity of the gaskets and the benzodiazepines. The contact angle for the BLACK JEA was 97.8 ± 1.7°, while the GRAY JEN elastomeric material exhibited a contact angle of 87.5 ± 0.8°. Based on these results, JEA gaskets have a more hydrophobic surface which can be attributed to their chemical nature. JEA is made of chlorobutyl rubber, while JEN consists of bromobutyl rubber. Both are derived synthetically from butyl rubber, a copolymer of isobutylene, but the subsequent halogenation is different, a chlorobutyl in JEA gaskets and bromobutyl in the JEN gaskets, according to the manufacturer (West Pharmaceuticals). The tensile strength has been found to be higher for composites containing bromobutyl rubber due to the strong polar character of the bromobutyl moiety [29]. Moreover, bromobutyl is more reactive than chlorobutyl, conferring it with a higher state of cure. Both rubber types have a phenolic resin cured to provide them with very high heat resistance, up to 120–170 °C [30].

The adsorption kinetic of both drugs on JEN showed a similar pattern characterised by a plateau from 15 to 180 min followed by a steady increase in the total amount of drug retained in the elastomeric gasket. This can be explained by complex equilibrium kinetics governed initially by a monolayer formation of drug molecules in solution adsorbed on the surface of the elastomeric material, followed by passive diffusion across the gasket. The monolayer capacity can be defined for physisorption as the amount needed to cover the surface with a complete monolayer of molecules in a close-packed array. The Freundlich isotherm can apply to this adsorption process as it occurs on a heterogeneous surface that dictates the exponential distribution of active sites and their energies [31]. The surface of the elastomeric gaskets at time 0 is not uniform, as observed in Figure 4, and different energy sites can dictate the physisorption between drug molecules in the solution and the chloro- and bromobutyl rubber from the elastomers. However, subsequently, drug molecules can diffuse inside the elastomer till the equilibrium is achieved over time. A dramatic increase in drug adsorbed at 36 months is found due to a loss in elastomer integrity, which allows a greater drug diffusion.

Surprisingly, the adsorption kinetics of the benzodiazepines for the JEA elastomers differed from the JEN gaskets. As mentioned above, the bromobutyl rubber of the JEN is more reactive than the chlorobutyl of the JEA, conferring it with a higher state of cure. We hypothesize that at earlier times, the resin coating layer from the JEA gaskets was solubilised with the methanol utilised to extract the benzodiazepines from the elastomers. This can explain why at 60 min, a significantly higher extracted drug from the JEA gaskets was found as the monolayer was disrupted. This also explains why the diffusion process is delayed, and within the 1000 min of the study drug retained in the elastomer did not increase steadily like in the JEN’s adsorption kinetics.

Based on these findings, we explored the amount of drug adsorbed on the surface of the gaskets at different times using NIR spectroscopy in more detail (Figure 8). Clear differences were observed between the JEA and the JEN elastomeric gaskets. After the second derivative, only a correlation was observed in the range between 1120 and 1250 nm in the JEA, while a linear correlation was found between the signal obtained from the surface of the elastomers and the total drug content retained in the JEN gaskets at different wavelengths (1210, 1230, 1380, 1397, and 1650 nm).

A PLSR model was developed to correlate the amount of drug retained in the gasket with the NIR spectroscopic signal obtained from the surface of the gasket (Figure 9). Surprisingly, a good linear correlation (R^2^ > 0.9) was found in the JEN elastomers, demonstrating the formation of the monolayer on the gasket surface. However, the RMSE was high, which can be related to the fact that at later time points, the passive diffusion across the elastomer governs the process rather than the monolayer formation on the surface of the gasket. In the case of the JEAs, the correlation found was much poorer, and only after removing the data points at 15, 30, and 60 min, was a trend found between NIR spectroscopic data and drug amount quantified by HPLC. This can corroborate the above-discussed hypothesis about the solubilisation of the resin coating layer of the JEAs, which disrupted the monolayer formation process.

## 4. Discussion

In this work, the stability of these types of formulations is defined as the time (t_90_) at which 90% of the initial drug concentration remained unchanged at 25 °C storage conditions, as this specification has been previously suggested by USP [32] for diazepam and midazolam solutions and in the works of Ezquer-Garin [33] and Mékidèche [34]. According to the t_90_ criteria as the definition of expired date, it is denoted as the limit of drug stability of the studied formulations. The stability data shown in Figure 5 shows that the midazolam formulation with sealing type PH 701/50 C BLACK (coded as MJEA) is not suitable. Moreover, the interaction between midazolam and the sealing type 4023/50 GRAY leads to a colour change, making this formulation, coded as MJEN, not suitable either.

In our experimental conditions, diazepam formulations were more stable than the midazolam formulations. Fyllingen et al. have reported that after 10 years of storage, less than 1.5% degradation was observed for diazepam injections [35]. Similar results have recently been reported by Itin et al. [36]. However, for midazolam, Gilliot et al. found that at 25 °C, its stability was compromised after 6 months [37]. Moreover, Feng et al. have reported that midazolam is degraded in acid media [38].

One possible explanation for the different stability behaviour between benzodiazepines can be due to the different pH of the formulations. Diazepam formulations are prepared based on the addition of propylene glycol, ethanol, and benzyl alcohol, which can solubilize the benzodiazepine at a neutral pH. However, for the midazolam formulations, benzodiazepine is used as a chlorhydrate salt with an acid pH. The ionisation of the benzodiazepine can be a key factor in the interaction of the benzodiazepines and the sealing materials.

Novel elastomeric bromobutyl gaskets (JEN) are suggested to reduce benzodiazepine adsorption and prevent drug loss in the solution. The compatibility of the injection of recombinant antitumor–antivirus protein and bromobutyl rubber stoppers has also been evaluated to assess the possible leaching of migrants from the stoppers to the injectable solutions. The leaching materials from bromobutyl stoppers were lower than the permitted daily exposure, indicating a safe profile for this type of elastomeric material [39]. The low leachable profile, along with the reduced benzodiazepine adsorption, makes bromobutyl elastomeric gaskets a good primary packing material for the preparation of benzodiazepine autoinjectors.

## 5. Conclusions

The selection of appropriate primary packing material is key to ensuring the physicochemical stability of benzodiazepine parenteral formulations, such as autoinjectors. Based on our results, bromobutyl elastomeric materials seem to be a better choice than the conventional chlorobutyl elastomeric gaskets due to the lower drug adsorption for the drugs studied in this article. Diazepam and midazolam parenteral formulations were stable (>90% drug content) at 4 °C over 12 months. However, drug loss was significantly higher at 25 °C when the chlorobutyl elastomeric gaskets were utilised.

## Figures and Tables

**Figure 1 pharmaceutics-14-02061-f001:**
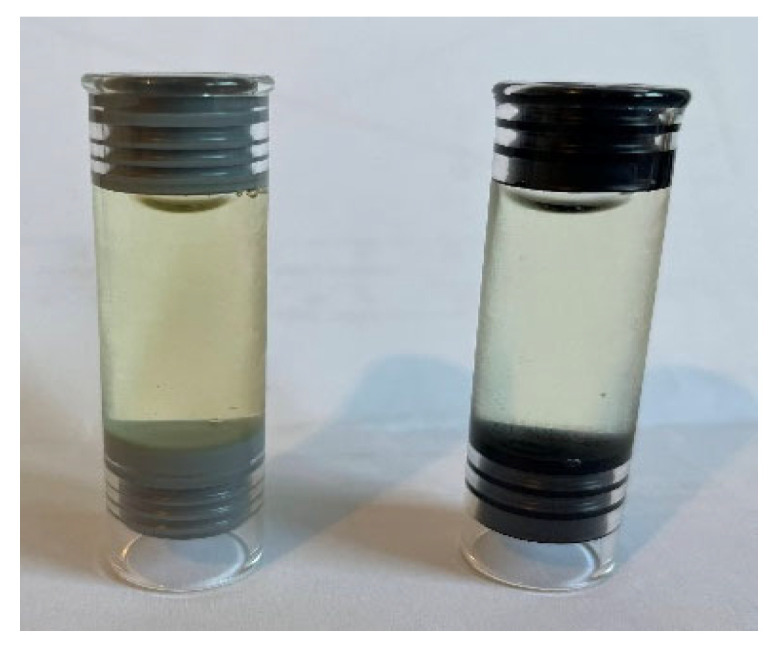
Example of the two sealing elastomeric gasket pieces (black colour right one and grey colour left part of the figure) located in the glass syringes.

**Figure 2 pharmaceutics-14-02061-f002:**
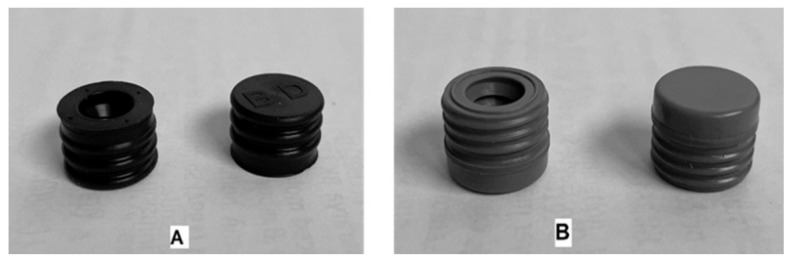
Pictures of the sealing pieces used in this study. Key, elastomer PH 701/50 C BLACK coded as JEA (**A**) and elastomer 4023/50 GRAY coded as JEN (**B**).

**Figure 3 pharmaceutics-14-02061-f003:**
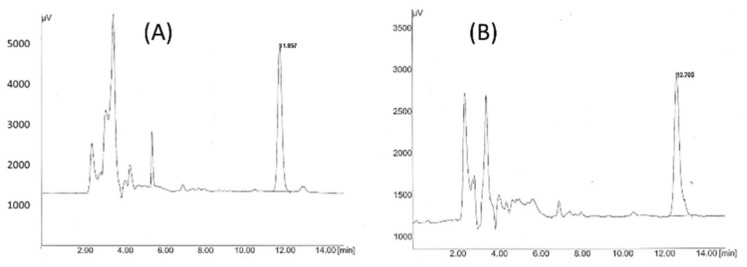
Chromatograms of diazepam (**A**) and midazolam (**B**) at 1.5 μg/mL. Retention times were 11.9 and 12.7 min, respectively.

**Figure 4 pharmaceutics-14-02061-f004:**
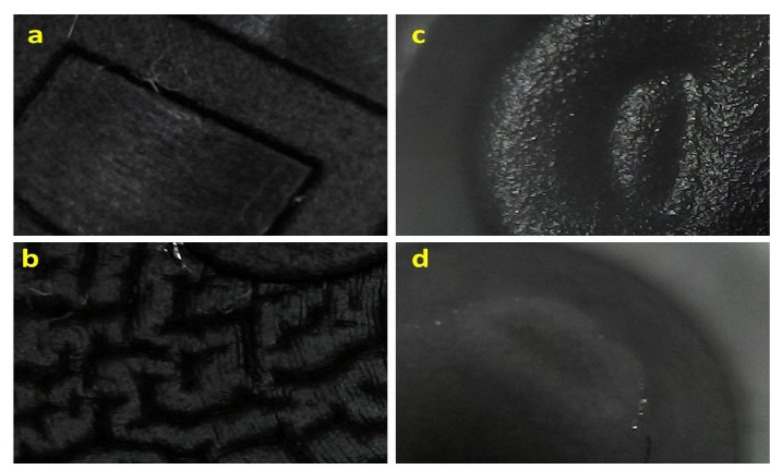
Micrographs obtained with a digital microscope at 200× magnification. Key: (**a**) JEA at time zero; (**b**) JEA after 36 months of exposition to the midazolam solution; (**c**) JEN at time zero; (**d**) JEN after 36 months of exposition to the midazolam solution.

**Figure 5 pharmaceutics-14-02061-f005:**
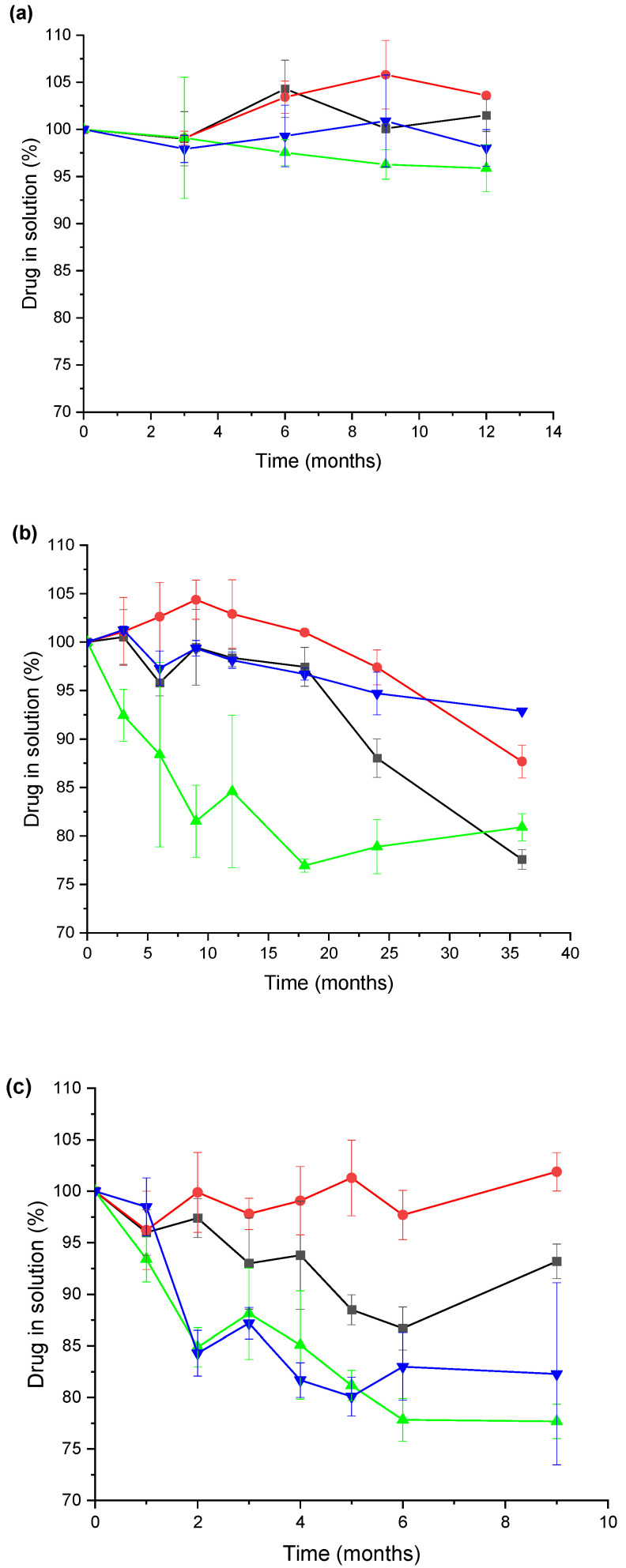
Drug recovery in solution under different storage conditions: (**a**) 4 °C, (**b**) 25 °C, and (**c**) 40 °C. Key: -■—DJEA, diazepam in sealing type PH 701/50 C BLACK, red; -●—DJEN, diazepam in sealing type 4023/50 GRAY; -⏶—MJEA, midazolam in sealing type PH 701/50 C BLACK; and -⏷—MJEN, midazolam in sealing type 4023/50 GRAY.

**Figure 6 pharmaceutics-14-02061-f006:**
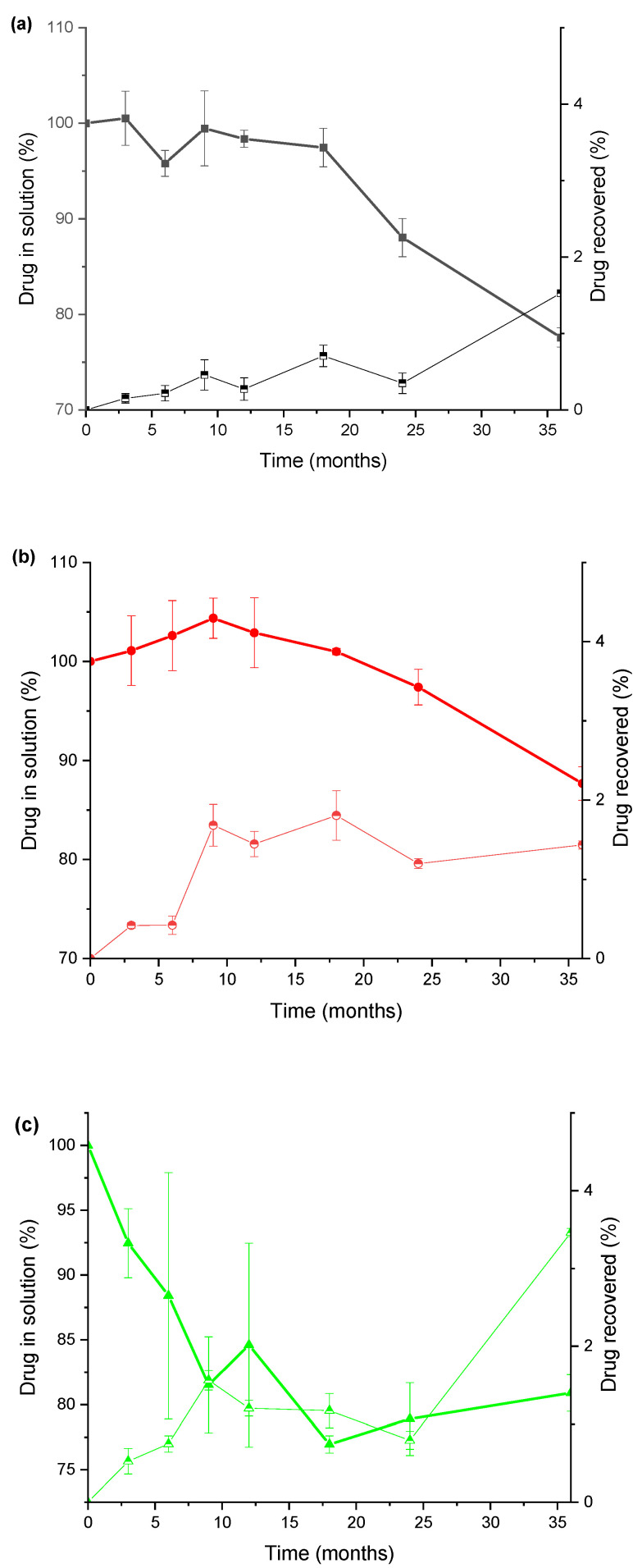
Correlation between drug recovery in solution (full colour symbol) and drug retained in the elastomeric gasket (half full colour symbol) at 25 °C. Key: (**a**) Diazepam in sealing type PH 701/50 C BLACK JEA, (**b**) diazepam in sealing type 4023/50 GRAY JEN, (**c**) midazolam in sealing type PH 701/50 C BLACK JEA, and (**d**) midazolam in sealing type 4023/50 GRAY JEN.

**Figure 7 pharmaceutics-14-02061-f007:**
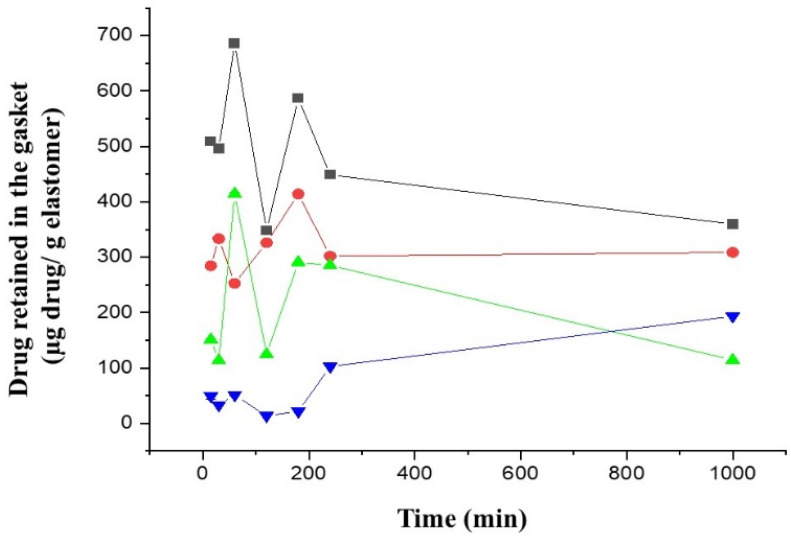
Adsorption of benzodiazepines on elastomeric gaskets. Key: -■—DJEA, diazepam in sealing type PH 701/50 C BLACK; -●—DJEN, diazepam in sealing type 4023/50 GRAY; -⏶—MJEA, midazolam in sealing type PH 701/50 C BLACK; and -⏷—MJEN, midazolam in sealing type 4023/50 GRAY.

**Figure 8 pharmaceutics-14-02061-f008:**
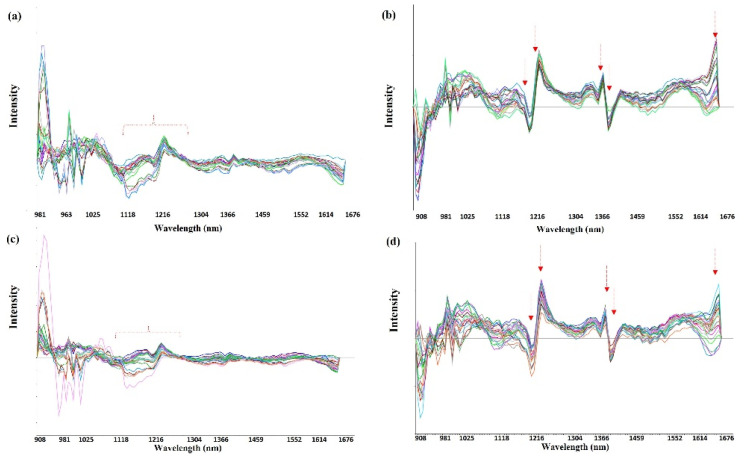
NIR accumulated spectra after data normalisation and a second derivative in the adsorption study for both drugs and two elastomeric materials. Key: (**a**) DJEA; (**b**) DJEN; (**c**) MJEA; and (**d**) MJEN.

**Figure 9 pharmaceutics-14-02061-f009:**
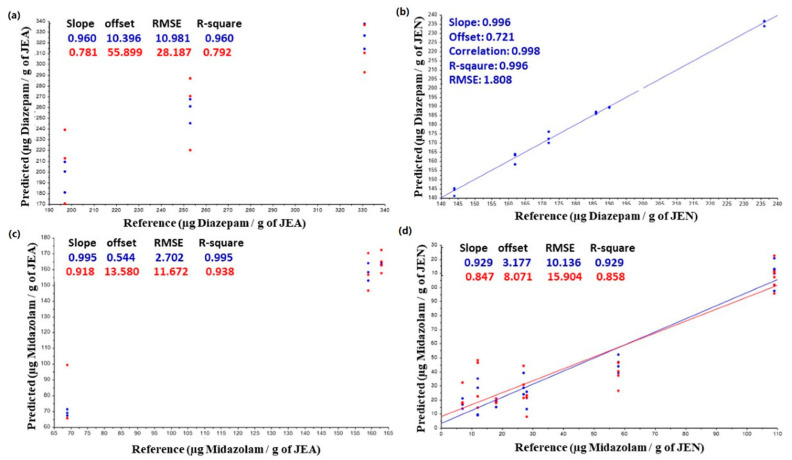
PLSR model to correlate NIR spectroscopic signal from the elastomer surface and drug retained in the gasket. Key: (**a**) DJEA; (**b**) DJEN; (**c**) MJEA; and (**d**) MJEN.

**Table 1 pharmaceutics-14-02061-t001:** Physicochemical characteristics of diazepam and midazolam.

	Diazepam	Midazolam
Chemical structure	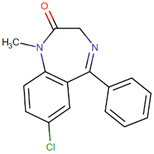	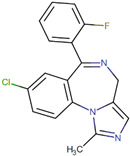
Molecular weight (g/mol)	284.7	325.8
pKa	3.4 (Source: MERCK INDEX (1996))	6.2 (Source: REMINGTON (2006))
Log P	2.82 (Source: SANGSTER (1994))	2.5 (Source: PubCHEM (2022)

**Table 2 pharmaceutics-14-02061-t002:** Composition of the diazepam and midazolam parenteral formulations.

Composition of the Diazepam		Composition of Midazolam	
Diazepam	2.5 g	Midazolam chloride	2.5 g
Ethanol 96°	42.6 mg	Sodium metabisulphite	0.5 g
Propylene glycol	233 g	Water	Up to 500 mL
Sodium benzoate	23.75 g		
Benzoic acid	1.25 g		
Benzylic Alcohol	7.83 g		
Water	Up to 500 mL		
Concentration solution pH solution (adjusted with benzoic acid)	0.3 g/100 mL 6.5	Concentration solution pH solution (adjusted with HCl 1M)	0.3 g/100 mL 2.9

**Table 3 pharmaceutics-14-02061-t003:** Conditions of the stability study.

Study	Storage Conditions	Minimum Time according to ICH	Time of the Study	Testing Frequency (Months)
Long term	25 ± 2 °C 60 ± 5% R.H.	12 months	36 months	0, 3, 6, 9, 12, 18, 24 and 36
Accelerated	40 ± 2 °C 75 ± 5% R.H.	6 months	9 months	1, 2, 3, 4, 5, 6, and 9
Refrigerated	4 ± 2 °C	6 months	12 months	3, 6, 9, and 12

**Table 4 pharmaceutics-14-02061-t004:** Visual appearance of the sealing elastomeric gaskets of the syringes in contact with the diazepam and midazolam solutions during the stability test study.

SEALING PIECES AT INITIAL OF STUDY	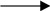	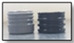	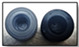
Conditions	Sampling month	Diazepam	Midazolam
4 ± 2 °C	3	-	-
6	-	-
9	-	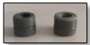
12	-	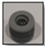
25 ± 2 °C/60 ± 5% R.H.	3	-	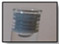
6	-	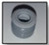
9	-	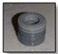
12	-	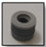
18	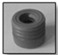	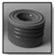
24	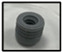	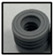 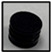
36	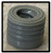	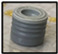 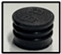
40 ± 2 °C/75 ± 5% R.H.	1	-	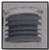
3	-	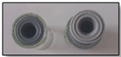
4	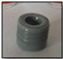	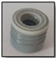
5	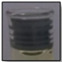	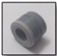
6	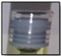	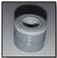
9	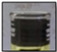	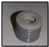

## Data Availability

Not applicable.

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
