# Peer review of "Effect of Primary Packaging Material on the Stability Characteristics of Diazepam and Midazolam Parenteral Formulations"

_pharmaceutics, 2022, doi:10.3390/pharmaceutics14102061_

Round 1

Reviewer 1 Report

In this work, The authors have packaged Diazepam and midazolam in glass syringes as parenteral solutions using two different elastomeric sealing materials. These Syringes were then stored at different temperatures and time points over 3 years; physical appearance, benzodiazepine sorption on the elastomeric sealing materials, and drug content in solution were assayed. The study indicates that the novel rubber material made of bromobutyl derivatives (4023/50 GRAY) is a better choice for manufacturing autoinjectors due to lower drug adsorption. They observed that Diazepam showed a better stability profile than midazolam. The amount of drug adsorbed on the surface of the elastomer was measured by NIR and correlated using chemometric models with the amount retained in the elastomeric gaskets quantified by HPLC.

The study is thorough and well designed; however, there are a few concerns that the authors should address.

  1. Authors should explain the basis of the selection of Testing frequency (months)-table 1.
  2. What is the author's opinion on using DVS (dynamic vapor sorption) to get absorption-desorption hysteresis for this kind of interaction study?
  3. In lines 360-361, the authors mentioned, “From a pharmacological point of view, midazolam has been reported by  McDonough et al., as more appropriate than diazepam as an anti-convulsant agent.” As both drugs are used for their anti-convulsant properties, there is no need to compare pharmacological points here.
  4. In section 2.6, the authors have not mentioned the equipment's make and model.
  5. Ref 27, page numbers are missing.

Author Response

The study is thorough and well designed; however, there are a few concerns that the authors should address:

  1. Authors should explain the basis of the selection of Testing frequency (months)-table 1.

We thank the reviewer for the comment. Currently, the regulations in Spain regarding drug stability studies follow the principles developed by the International Committee for Harmonization (ICH), established in the following main guideline, ICH Q1A(R2) "Relative to the procedures of the stability studies of new active ingredients and derivative drugs".

This was our starting point, but later we decided to add more sampling points for a better understanding of the degradation process of the active ingredient.

  1. What is the author's opinion on using DVS (dynamic vapor sorption) to get absorption-desorption hysteresis for this kind of interaction study?

It is a very interacting method but in our experimental conditions for liquid aqueous solutions contact angle is a more relevant technique to study water physicochemical interactions with the elastomeric gaskets.

  1. In lines 360-361, the authors mentioned, “From a pharmacological point of view, midazolam has been reported by McDonough et al., as more appropriate than diazepam as an anti-convulsant agent.” As both drugs are used for their anti-convulsant properties, there is no need to compare pharmacological points here.

 We agree with the reviewer, and we have removed it from the text.

  1. In section 2.6, the authors have not mentioned the equipment's make and model.

We agree with the reviewer and now it has been included in the new version of the manuscript.

Measurements were made with Image J software (Pendent drop plugin), which is a public domain, Java-based image processing software package.

  1. Ref 27, page numbers are missing.

The are no page numbers in this article as shown in the internet address: https://pubmed.ncbi.nlm.nih.gov/30961131/

Reviewer 2 Report

Parenteral formulations require special care in terms of purity, API stability and the possibility of API interacting with the packaging material. The authors of the study focused on the stability of diazepam and midazolam in glass syringes using two different elastomeric sealing materials. On the basis of research, they conclude that which of the materials used is better suited for a given form of the drug.

The research proposed in this paper seems to be useful, however, whether rubber material made of bromobutyl derivatives is suitable for contact with organic solvents, I have doubts. There are no studies confirming the microbiological purity of the obtained drugs in the conducted studies. In HPLC tests, it is also necessary to perform API purity tests using the HPLC method, the more so that the chromatograms show additional peaks that may indicate the formation of decomposition products.

Author Response

The research proposed in this paper seems to be useful, however, whether rubber material made of bromobutyl derivatives is suitable for contact with organic solvents, I have doubts. There are no studies confirming the microbiological purity of the obtained drugs in the conducted studies. In HPLC tests, it is also necessary to perform API purity tests using the HPLC method, the more so that the chromatograms show additional peaks that may indicate the formation of decomposition products.

We thank the appreciation for the reviewer, but we have focused our work to the interaction between drug formulations and the sealing materials available in the market. The possible interaction between the rubber material made of bromobutyl derivatives in contact with the organic solvent of the diazepam formulation is one of the questions to be studied in this work. Formulations were not prepared in aseptically conditions and hence, microbiological studies were not performed. APIs were bought with a high purity (≥ 99%). No additional peaks were observed with any of the raw materials.

Reviewer 3 Report

The article entitled “Effect of primary packaging material on the stability characteristics of diazepam and midazolam parenteral formulations” address an interesting topic. This field of research has been widely studied, strengthening that there is a need to provide solutions to the issue of drug sorption.

However, there are several drawbacks in the proposed version of the manuscript, and I recommend to resubmit the manuscript after having applied major revisions. Overall, to me, what is missing the most is data explaining the physico-chemical features of the drug, the inclusion of other experiments to ease the comparison of the results and help to explain them. Indeed, (too) many hypotheses can help to support the results, as it is also stated by the authors in their conclusion (l 354-359).

I recommend to change and/or add the following:

-          The structure, the pKa(s) and the log D (log P as a function of pH) of the two drugs should be added

-          The word “stability” is not appropriate to me: the amounts of drugs are followed up, not their stability/degradation. Indeed, the methods used are not proved to be stability indicating. The word assays and/or recovery should be used instead of stability

-          The accuracy of the liquid chromatography method and its relevance to the context (e.g. are accuracy, precision… appropriate do the context?) should be discussed. An accuracy profile would be interesting

-          The composition of the diazepam and midazolam (2.2) should be reported in a table ; please provide the final concentration and the pH of the solutions

-          Results and discussion should be divided in several parts (e.g. subtitles) to ease the reading: indeed I sometimes wondered what was the aim of the information provided. A good outline of this part of the manuscript is necessary to understand why some results are provided

-          A discussion on the results is missing. Clearly, some data exists on the interaction of organic compounds with chlorobutyl and chlorobutyl casks: when typing the keywords : “adsorption bromobutyl chlorobutyl casks” I found some studies. It would help to perceive furthermore how much these results may apply to other context (e.g. for other compounds, vehicles, concentration…) which is very important.

-          Other tests should be performed to check the influence of the change from chloride containing containers to bromide ones. For instance, experiments could be performed so as to add information on figure 7: for instance, if the vehicles of diazepam are swiped with that of midazolam, are the results similar? If the pH are changed are the results similar? Clearly this is important to assess if the sorption of drugs is “elastomeric gaskets” and/or “drug” and/or “vehicle”and/or”pH” dependant. Thus, the authors could, at least for these drugs, really assess as stated in the title the “Effect of primary packaging material”

Round 2

Reviewer 3 Report

To me the manuscript is almost suitable for publication. One last recommandation : please moderate your conclusions were revelant. For instance : "Bromobutyl elastomeric materials are a better choice than the conventional chlorobutyl elastomeric gaskets due to a lower drug adsorption." should be changed for instance for: "Based on our results, bromobutyl elastomeric materials seem to be a better choice than the conventional chlorobutyl elastomeric gaskets due the lower drug adsorption for the drugs studied in this article."